# Production of hydrogen from offshore wind in China and cost-competitive supply to Japan

Shaojie Song [1], Haiyang Lin[1,2 ✉], Peter Sherman [3], Xi Yang [1,4], Chris P. Nielsen[1], Xinyu Chen[1,5] & Michael B. McElroy [1,3 ✉]

The Japanese government has announced a commitment to net-zero greenhouse gas emissions by 2050. It envisages an important role for hydrogen in the nation's future energy economy. This paper explores the possibility that a significant source for this hydrogen could be produced by electrolysis fueled by power generated from offshore wind in China. Hydrogen could be delivered to Japan either as liquid, or bound to a chemical carrier such as toluene, or as a component of ammonia. The paper presents an analysis of factors determining the ultimate cost for this hydrogen, including expenses for production, storage, conversion, transport, and treatment at the destination. It concludes that the Chinese source could be delivered at a volume and cost consistent with Japan's idealized future projections.

[1] John A. Paulson School of Engineering and Applied Sciences, Harvard University, Cambridge, MA 02138, USA. [2] Institute of Thermal Science and Technology, Shandong University, Jinan 250061, China. [3] Department of Earth and Planetary Sciences, Harvard University, Cambridge, MA 02138, USA. [4] School of Economics and Management, China University of Petroleum Beijing, Beijing 102249, China. [5] State Key Laboratory of Advanced Electromagnetic Engineering and Technology, School of Electrical and Electronic Engineering, Huazhong University of Science and Technology, Wuhan 430074, China. ✉email: haiyanglin@seas.harvard.edu; mbm@seas.harvard.edu

Japan has recently announced its Green Growth Strategy to transition to a net-zero greenhouse gas emission economy by 2050[1]. This is a formidable challenge. Fossil fuels currently account for 87% of Japan's primary energy consumption[2], responsible for annual emissions of 1.1 gigatons (Gt) of $CO_2$, broken down as follows: electricity generation 43%, industry 28%, transportation 19%, and others 10%[3]. Notably, more than 95% of fossil fuel consumption in Japan is supplied by imports[2,4].

Japan's plans for its low-carbon energy future envisage an important role for hydrogen[1]. The main types of hydrogen include gray, black, blue, and green[5]. Currently, hydrogen is produced globally primarily from natural gas and this supply is classified as gray. Production of 1 kg of gray hydrogen is associated with an emission of ~10 kg of $CO_2$. Hydrogen in China is mainly produced from coal which is classified as black. If the by-product $CO_2$ is captured and sequestered, the hydrogen is defined as blue. Hydrogen produced by electrolysis of water powered by renewables such as wind and solar is classified as green[6]. Production of blue and green hydrogen accounted for less than 1% of the total global source of hydrogen in 2019[7]. The current annual demand for hydrogen in Japan amounts to 1.3 megatons (Mt) and is consumed primarily by the industrial sector including oil refining and production of ammonia and petrochemicals[8].

An ambitious target of 20 Mt year$^{-1}$ for low-carbon hydrogen consumption by 2050 has been set by the Japanese government. This demand corresponds to ~20% of the total final energy consumption in 2050 as estimated by Japanese research institutes[8,9]. Hydrogen is anticipated to play an important role in decarbonizing electricity generation and the hard-to-electrify sectors including heavy industry and long-distance transport and to substitute for fossil fuels as a zero-carbon feedstock in industrial sectors such as iron and steel. An interim target of 3 Mt year$^{-1}$ for hydrogen consumption by 2030 is identified in Japan's government report[1].

An abundant and low-cost supply of hydrogen is an indispensable requirement in achieving the goal for a viable future hydrogen economy. Japan has set price targets for the supply of low-carbon hydrogen at $3 kg$^{-1}$ by 2030, declining to $2 kg$^{-1}$ by 2050[10]. The ultimate objective is to have the price for hydrogen competitive with projected future costs for natural gas, the latter estimated at about $10 per million British thermal units. The price for hydrogen on an energy equivalent basis would amount to about $1.4 kg$^{-1}$ in this case. The price of $2 kg$^{-1}$ targeted for 2050 assumes a premium to allow for the environmental benefits of hydrogen or equivalently a penalty for the $CO_2$ emissions associated with the alternative consumption of natural gas.

The domestic production of green hydrogen from renewable energy in Japan is projected by the International Energy Agency to cost about $6 kg$^{-1}$ for 2030, considerably higher than the corresponding target of $3 kg$^{-1}$, reflecting the high costs of domestic renewable electricity generation[7]. Thus, Japan's vision for its future hydrogen economy projects a significant supply from imports. Hydrogen could be shipped to Japan in a variety of possible forms: as liquid hydrogen, as a component of ammonia, or bound to a chemical carrier such as toluene[1]. In the case of supplies converted to ammonia, the hydrogen could be recovered prior to its eventual use in Japan or alternatively the ammonia could be employed directly as a zero-carbon energy source or chemical feedstock[11]. In the case of hydrogen bound to toluene, forming methylcyclohexane (MCH), the hydrogen could be stripped from the carrier for deployment at the destination and the toluene could be returned to its point of origin to facilitate an additional supply[12]. Potential foreign sources include supplies from among others Australia, New Zealand, Brunei, Saudi Arabia, and Norway, with projected import costs in 2030 of $3–4 and $3–7 kg$^{-1}$, respectively, for green and blue hydrogen[7,13]. In summary, the supplies of hydrogen from currently anticipated sources are relatively expensive compared to the nation's targets.

In this work, we perform a detailed techno-economic analysis of the potential for a green hydrogen supply chain to Japan delivered from offshore wind produced in China on an hourly basis from every Chinese coastal province, considering several possible wind investment levels, electrolysis technologies, and transport mechanisms. We estimate the quantity and cost of green hydrogen that could be produced in China and delivered to Japan for the target years (2030 and 2050) highlighted in current Japanese projections. We find that the Chinese offshore source could be delivered at a volume and cost consistent with Japan's idealized future projections.

## Results

**Projection of a China–Japan hydrogen supply chain.** We choose to emphasize the potential source of green hydrogen that could be produced from offshore wind in China. A prior analysis suggests that electricity produced from this source could be competitive on a cost basis with existing sources of power from nuclear or even coal before 2030[14]. The offshore wind resource in China could provide potentially as much as 12 petawatt hours of electricity annually, approximately four times the demand for wind power projected nationally for 2050[14,15]. The advantage of this offshore source for China relates to the extensive range of environments with water depths less than 60 meters in the nation's exclusive economic zone, which contributes to a significant reduction in costs for the production of wind power (Supplementary Fig. 4). In contrast, economically viable offshore wind resources are more limited for Japan, reflecting the more rapid increase of water depths as a function of distance from shore.

We consider three specific technologies for water electrolysis, alkaline electrolyzer cells (AEC), proton exchange membrane electrolyzer cells (PEMEC), and solid oxide electrolyzer cells (SOEC), including estimates of how costs for these devices might be expected to vary in the future[7]. We estimate the cost for hydrogen delivered to Japan from the three potential transport mechanisms: liquid hydrogen, MCH, and ammonia. The analysis allows for expenses associated with the production of the hydrogen carriers and the construction and operation of reservoirs for storage compensating for times when the wind-fueled source might be temporally low. The analysis allows also for expenses associated with storage at ports prior to shipping, costs for transport and costs for reconstitution of hydrogen from the ammonia and toluene supplies once delivered to the Japanese destination. The expense for the release of hydrogen (i.e., dehydrogenation) from MCH and ammonia depends on the energy source employed. We consider three possible sources: economically advantaged waste heat from industrial sources; combustion of hydrogen or ammonia; and combustion of natural gas. Japan's road map for its hydrogen future envisages the development of a dehydrogenation technology using waste heat, projecting that the first such system should be commercially available by 2030[10]. We consider separately the costs associated with the delivery of ammonia without dehydrogenation at the destination, recognizing that ammonia could be employed directly as a fuel in power generation.

In the baseline scenario, we assume moderate reductions of the costs of offshore wind projects in the future decades as estimated by the National Renewable Energy Laboratory (NREL). The AEC system is considered in the baseline scenario since it is technically mature and the most cost favorable option for water electrolysis. Pressurized tanks are assumed as the default reservoir for storage. The economically advantaged industrial waste heat is assumed as the energy source for the release of hydrogen. The other scenarios

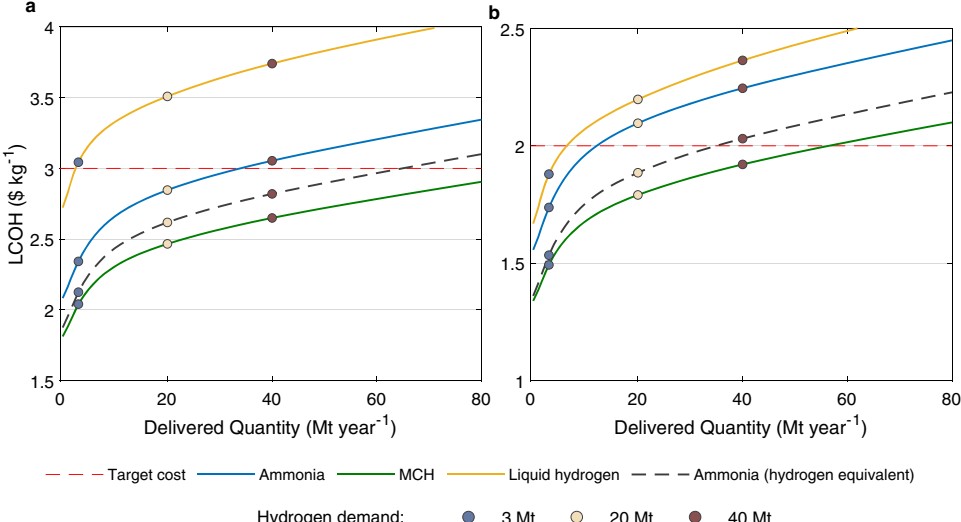

**Fig. 1 Supply curves for hydrogen produced from offshore wind in China and delivered to Japan. a** and **b** are for 2030 and 2050, respectively. Results from the baseline scenario are presented. The levelized cost of hydrogen (LCOH) is calculated as the weighted average cost for the cumulative quantity of hydrogen. Data for the three transport mechanisms are represented by solid curves with different colors. Data for delivery of ammonia without dehydrogenation at the destination are shown in gray dashed curves. Projections of hydrogen demand in Japan are tagged by colored circles. The targeted hydrogen costs for Japan are indicated with red dashed lines. MCH represents the hydrogen transport mechanism as in methylcyclohexane.

assume different costs of offshore wind projects or different options in the hydrogen supply chain. For example, the cost savings for hydrogen storage that could be realized with the use of salt caverns if available for geological storage is investigated separately.

**Large amount of cost-competitive hydrogen from China**. The potential quantity of hydrogen that could be delivered from the Chinese offshore wind source to Japan for 2030 and 2050 is illustrated in Fig. 1 as a function of the related levelized costs of hydrogen (LCOH). This figure presents results from the baseline scenario (see Methods for its assumptions). For delivery of either a given amount of hydrogen or hydrogen-equivalent ammonia, the MCH mechanism is identified as the least expensive pathway, followed, in ascending order by ammonia, ammonia (with dehydrogenation), and liquid hydrogen (Fig. 1). The LCOH in the MCH case could be as low as $2.0 and $1.8 kg$^{-1}$, respectively while meeting the demand levels of 3 and 20 Mt year$^{-1}$ targeted by the Japanese government for 2030 and 2050. Notably, these values of LCOH are significantly lower than the prices targeted for these years by the Japanese government, $3 and $2 kg$^{-1}$, respectively. The LCOH for the three transport mechanisms could be reduced further if the salt cavern option were available for on-site storage in China (Supplementary Fig. 2). This choice would be particularly beneficial in the case of liquid hydrogen, allowing it to become more competitive than ammonia for 2050.

The present analysis suggests that China could supply a source of cost-competitive green hydrogen even higher than the quantities, 20 Mt year$^{-1}$, envisaged in the Japanese target for 2050. Current Japanese policy, as elaborated in the Green Growth Strategy, contemplates potential hydrogen demands by 2050 of 6, 7, and 5–10 Mt year$^{-1}$ in the transportation, industry, and power sectors respectively[1]. Power plants using hydrogen or ammonia are anticipated to account for no more than 10% of the future electricity generation. The balance would come from renewable sources (50–60%, including wind, solar, hydro, biomass, and geothermal) and from a combination of nuclear and thermal plants (30–40%), the latter equipped with carbon capture[1]. Considering a more ambitious goal for use of hydrogen to replace

the envisaged nuclear and thermal components of the power sector[16], future demand for hydrogen could reach as high as 40 Mt year$^{-1}$. The data in Fig. 1b indicate that this demand could be accommodated by hydrogen from China in 2050 at a cost of less than $2 kg$^{-1}$. Even when waste heat is unavailable in the future, this demand could still be satisfied by the supply and direct use of ammonia (Supplementary Fig. 3).

The geographic distribution of LCOH for the MCH transport mechanism is displayed in Fig. 2 (a for 2030 and b for 2050). The costs with feasible locations for China's offshore wind range from less than $2 kg$^{-1}$ to more than $6 kg$^{-1}$. Mean capacity factors, distances to shore, and water depths are recognized as the important considerations determining the LCOH for each offshore location (Supplementary Fig. 4). Interannual variability of the wind source may lead to deviations with respect to the 30-year period mean, as illustrated by the year-by-year variability in the nationally aggregated supply (Supplementary Fig. 5), with notable provincial examples being Fujian and Guangdong. The most favorable cost-competitive source is identified with Fujian, followed by important additional contributions from Liaoning, Zhejiang, and Shandong (Table 1). Notably, the quantity of hydrogen available from a single province, such as Fujian, could satisfy the entirety of the future demand projected for Japan.

The cost breakdown of LCOH for delivery of the potential products to Japan is shown in Fig. 3. Cost for hydrogen production represents the largest contribution to the overall expense, accounting for 47–70% of the total. The conversion process, including hydrogenation and where necessary dehydrogenation, is responsible for 19–35%, of the total cost. Storage and transport are minor contributors, especially for the former if the geological storage opportunity is available, and for the latter reflecting the relatively short distances separating the Chinese and Japanese ports of origin and delivery (Supplementary Table 5). The sensitivities of LCOH to the specific assumptions adopted with respect to the future reductions in offshore wind farms and electrolysis technologies are presented in Tables S6 and S7. The results suggest that this Chinese source could supply cost-competitive hydrogen to Japan for 2030 even if the offshore wind deployment is projected to follow the high-cost scenario. The choice of other water electrolysis systems (PEMEC or SOEC)

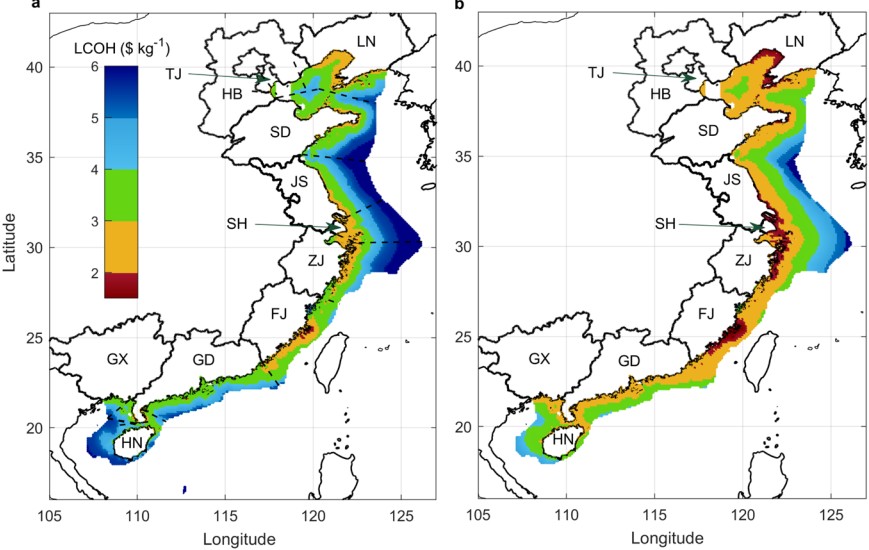

**Fig. 2 Geographic distribution of levelized cost of hydrogen (LCOH) estimated for offshore China. a** and **b** are for 2030 and 2050, respectively. Results from the baseline scenario assuming the methylcyclohexane (MCH) transport mechanism are presented. The dashed black lines denote offshore wind areas which are in closest proximity to particular provincial-level administrative divisions. LN, TJ, HB, SD, JS, SH, ZJ, FJ, GD, GX, and HN are the abbreviations for Liaoning, Tianjin, Hebei, Shandong, Jiangsu, Shanghai, Zhejiang, Fujian, Guangdong, Guangxi, and Hainan, respectively.

**Table 1 Cost-competitive hydrogen potential from coastal provinces or municipalities in China.**

| Province or municipality | Abbreviation | 2030 | | 2050 | |
|---|---|---|---|---|---|
| | | Mean | SD | Mean | SD |
| Fujian | FJ | 28.2 | 6.0 | 21.2 | 6.9 |
| Liaoning | LN | 16.9 | 4.1 | 11.2 | 4.2 |
| Zhejiang | ZJ | 15.8 | 4.7 | 10.1 | 4.9 |
| Shandong | SD | 8.6 | 4.5 | 3.0 | 2.5 |
| Shanghai | SH | 7.7 | 1.9 | 5.8 | 2.2 |
| Jiangsu | JS | 6.8 | 2.7 | 3.5 | 2.0 |
| Guangdong | GD | 6.6 | 6.0 | 0.15 | 2.6 |

The cost-competitive potential (Mt year⁻¹) is defined as the maximum quantity of hydrogen that could be delivered to Japan at the levelized cost of hydrogen (LCOH) of \$3 kg⁻¹ in 2030 and \$2 kg⁻¹ in 2050. Results from the baseline scenario are presented assuming the methylcyclohexane (MCH) transport mechanism. Data were calculated using the MERRA-2 wind field data for 30 years and the standard deviation (SD) represents interannual climate variability. The potentials for the remaining provinces and municipalities are less than 1 Mt year⁻¹ and thus not shown.

would lead to minor increases in LCOH which still meet the targets envisaged by the Japanese government.

**Implications for domestic and international policies**. The overall analysis suggests that Chinese offshore wind could provide an important and cost-competitive source of green hydrogen to Japan. The estimated LCOH from this Chinese source is lower than the costs for green or blue hydrogen associated with either domestic production or imports from more remote regions such as Australia, Norway, or the Middle East[7]. Japan currently relies heavily on imports of fossil fuels from the Middle East and Australia. The transition to a low-carbon hydrogen emphasis energy future recognizing the diversity of potential sources including though not necessarily confined to the source from China would allow Japan to advance the goal for its overall "3E + S" ambition, the plan referring to a combination of objectives for Energy security, Economic affordability, Environment, and Safety.

The export of low-cost green hydrogen from China to Japan could contribute to an important additional trade relationship

between the two countries. We assumed implicitly that the offshore power consumed in the hydrogen production process would be financed by Chinese sources with a similar assumption for the activities involved in the production and transport of this hydrogen. An alternate approach could involve cooperative financing for all these functions facilitated by a combination of Chinese and Japanese investors. Japan could take a lead in the shipping of the hydrogen produced from the Chinese source. We would note in this context that Japan aspires to play an important role in a hypothetical future global hydrogen economy by the development of the liquid hydrogen ship launched in 2019[17].

Japan has plans to invest significantly in fuel cell technology for use not only in transportation but also in its residential sector[10]. A closer commercial relationship between China and Japan could facilitate comparable applications of these technologies in China. Deployments of green hydrogen in the industrial and power sectors as envisaged for Japan could allow for similar investments in China, allowing both countries to advance their overall plans for future carbon neutrality.

Finally, we would note that the quantities of offshore power potentially available from wind in China are more than sufficient to satisfy projected future demands for power and hydrogen in China in addition to the source envisaged here for Japan which would amount to less than 10% of the available total. Here we emphasize opportunities for the supply of green hydrogen from China to Japan. A comparable market could exist for Chinese hydrogen in the Republic of Korea, which also has plans for significant investments in hydrogen, handicapped similarly to Japan in terms of the limited resources available for the development of domestic sources of renewable power[18].

## Methods

**Overview of the supply chain scheme**. The China–Japan green hydrogen supply chain consists of hydrogen production, conversion, transport, and release (Supplementary Fig. 1). Power for the water electrolysis system is supplied from offshore wind resources in China. The hydrogen supply chain scheme is conducted on an hourly basis for every coastal province under all possible wind investment levels. Wind resource data for the 30-year interval (1990 to 2019) were used to account for climate variability. Three technologies for water electrolysis, AEC, PEMEC, and SOEC, are considered. The electricity produced by offshore wind turbines is assumed to be transported by cable to shore, where centralized electrolyzers are placed to produce hydrogen. An option, which is not explicitly considered here,

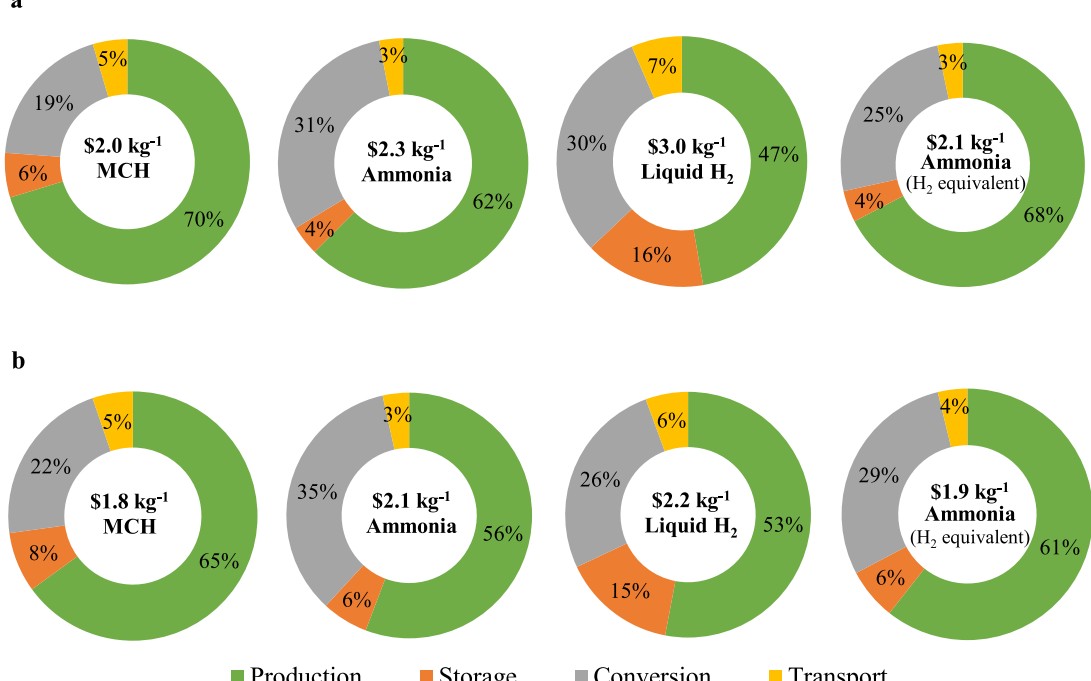

**Fig. 3 Cost breakdown of levelized cost of hydrogen (LCOH) for delivery of the potential products to Japan. a** and **b** are for 2030 (3 Mt of hydrogen) and 2050 (20 Mt of hydrogen), respectively. The costs are mean values calculated according to offshore wind conditions over the 30-year period. Results from the baseline scenario are presented. Conversion indicates expenses were appropriate for hydrogen compression, liquefaction, ammonia synthesis, and for hydrogenation and dehydrogenation of the chemical carriers. Transport defines costs for capital and energy included in shipping and pumping. Individual items are indicated by colors as defined in the legend. Expenses for electricity are included in each item based on its corresponding consumption. MCH represents the hydrogen transport mechanism as in methylcyclohexane.

could involve the production of hydrogen with electrolyzers sited at the wind farm rather than onshore. This could be appropriate for an offshore wind farm installed specifically to produce hydrogen. It is not however likely to be appropriate for the situation envisaged here, where the primary product of offshore wind is envisaged as the supply of carbon-free power for coastal China.

The volumetric energy density of hydrogen, 0.003 kWh $L^{-1}$ under ambient conditions, is extremely low compared to fossil fuels. Therefore, after its generation by electrolysis, hydrogen must be treated or adjusted to enable large-scale transport and storage to proceed economically. Liquefaction, chemical bonding to liquid organic hydrogen carriers (LOHCs), and synthesis of ammonia with pure nitrogen are considered as three hydrogen conversion options. Potential LOHCs include toluene, methanol, dibenzyltoluene, etc.[12]. Toluene is chosen in the present analysis because it is a commercially available chemical and this LOHC pathway has been used in a demonstration project transporting hydrogen from Brunei to Japan[19]. Toluene is reacted with hydrogen to produce methylcyclohexane (MCH).

Responding to the variability of the power source, hydrogen is stored in a temporary reservoir allowing delivery of a reliable supply to the selected conversion plant. Pressurized tanks are taken as the default option for gaseous hydrogen storage. The possibility of storage in a geological reservoir such as a salt cavern is considered separately. Storage of potential carrier options (liquid $H_2$, MCH, and ammonia), which is much cheaper than storage of the original gaseous product in pressurized tanks, is invoked to compensate for times when the wind-fueled source may be temporally low.

A further buffer storage system is deployed at the port to facilitate transfer to the ship engaged for transport of the product to Japan. On arrival in the Japanese port, an additional plant is employed to produce gaseous hydrogen from the carrier materials, indicated as hydrogen conversion II in Supplementary Fig. 1. Possible sources of heat for processing dehydrogenation includes waste heat, consumption of natural gas, and burning hydrogen. The unloaded materials (notably in this study toluene) are transported back to the port of origin.

To address the baseline economic analysis of hydrogen supply chains, the moderate cost for wind turbines, AEC for hydrogen production, and waste heat for dehydrogenation are the selected options in the baseline scenario. For comparison, the analysis summarized here also includes the sensitivity of results to the specific assumptions adopted with respect to wind turbine cost, electrolyzer type, and the heat sources for hydrogen release.

We normalize currencies to US dollars and all values to real monetary values for the year 2020[20]. The interest rate for economic analysis is assumed to be 7%. The electricity prices from the grid in China and Japan are assumed to be $0.1 and $0.16 k $Wh^{-1}$, respectively, equal to their current industrial tariffs.

The key steps for the production of hydrogen and for its delivery to Japan are described in the following sections.

**Offshore wind power generation**. The wind field data were taken from version 2 of the Modern-Era Retrospective analysis for Research and Applications (MERRA-2) of NASA's Goddard Earth Observing System[21]. Winds included in this dataset were obtained by retrospective analysis of global meteorological data using a state-of-the-art weather/climate model incorporating a variety of observational inputs from surface stations, aircraft, balloons, ships, buoys, dropsondes, and satellites. Reanalysis data for the 30-year interval (1990 to 2019) were used to account for interannual variability. Hourly wind speeds were provided by MERRA-2 at a horizontal resolution of 0.625° longitude by 0.5° latitude (equivalent to about 50 km by 56 km at mid-latitudes).

We calculated wind power generation on an hourly basis following Sherman et al.[14] using the power curve for the MHI Vestas Offshore Wind's V164-8.0 MW® turbine, a representative system used currently for offshore applications. Technical parameters for this type of wind turbine were described in ref. [22]. We considered further the potential applications of the MHI Vestas Offshore Wind's V164-9.5 MW® turbines introduced recently to the market. We found that the 9.5 MW option could increase the potential source of offshore wind power by about 19%. Assessment of the implications of the 9.5 MW option for the related levelized costs of hydrogen (LCOH) would require additional data on the relevant costs for the production and deployment of these machines. The future market could involve the deployment of even larger turbines as great as 15–20 MW[23]. This and other possible future systems should be assessed as they are proposed to enter the market. We would point out that the larger diameter of these larger turbines will require expanded spacing between individual turbines in order to avoid turbine-turbine interference.

Locations for offshore wind farms were restricted to China's Exclusive Economic Zone (EEZ) and to regions with water depths less than or equal to 60 m. The EEZ shapefile information and the water depth data were obtained respectively from the Maritime Boundaries Geodatabase[24] and from the one arc-minute gridded data available in the General Bathymetric Chart of the Oceans (GEBCO)[25]. The spatial data used in this study were converted and applied to grid cells with a horizontal resolution of 0.1° longitude by 0.1° latitude (about 8 km by 10 km at mid-latitudes). Offshore areas designated as "Special Marine Reserves" (environmentally protected regions) and shipping routes were excluded from the potential resource locations, as discussed in more detail in ref. [14]. The restriction on water depths was imposed in the present analysis since we elected to consider only

the potential for fixed-bottom wind turbines. Recent developments exploring the possibility of floating turbines could allow for deployment in deeper water environments (depths in excess of 60 m) but would entail significantly higher costs both for installation and maintenance[26]. As indicated earlier, the waters over much of China's EEZ are relatively shallow (<60 m, as shown in Supplementary Fig. 4b). We assumed a spacing between individual wind turbines of $7 \times 7$ rotor diameters (1.04 km$^2$) following the suggestion by Musial et al.[27]. The potential installed offshore wind capacity for each grid cell was calculated by multiplying the turbine nameplate capacity (8 MW in this case) by the number of turbines that could fit maximally into an individual cell. Power losses (on the order of several percent of the total generation) due to the downstream wake effect were not accounted for here since this phenomenon occurs on a scale too small to be simulated accurately using the MERRA-2 wind field[14,27].

The average total installed cost for the offshore wind power farms under construction and approved in China in 2018 was around $2700 kW$^{-1}$ (ref. [28]). In this study, we considered three future capital cost scenarios (high, moderate, and low) for offshore wind deployments based on three technology innovation scenarios (conservative, moderate, and advanced) discussed by the National Renewable Energy Laboratory (NREL) in its 2020 Annual Technology Baseline (ATB)[29]. The reductions in capital costs between 2018 and 2030 were projected at 49, 37, and 16% for the high, moderate, and low scenarios respectively. From 2030 to 2050, estimated capital costs were reduced further by 30, 24, and 11% for the three scenarios. The projected baseline capital costs were adjusted then spatially in each grid cell for the potential offshore areas. The effects on capital costs of water depth and distance from shore were estimated by a linear regression model developed by Sherman et al.[14]. This linear method represented an approximation of the results from the NREL offshore balance-of-system model[30], which included a detailed analysis accounting for the relationship between these two primary variables (water depth and distance from shore) and the costs associated with substructure and foundation, electrical infrastructure, and installation. Annual O&M costs were assumed to be 2% of capital costs[26], and the lifetime of offshore wind projects was assumed to be 20 years[31].

**Water electrolysis**. The techno-economic characteristics of the three specific technology options, AEC, PEMEC, and SOEC, are summarized in Supplementary Table 1. Capital costs include expenses for the electrolyzer, foundation, turn-key installation, and all necessary auxiliary systems such as feed water treatment and devices for cooling and purification. Annual O&M expenses include maintenance, stack replacement, and labor costs are expressed as a percentage of initial capital investments[32]. The AEC system is currently the most mature and durable among these three options. Chinese manufacturers consider $200 k W$^{-1}$ a realistic estimate for the contemporary cost of AEC, up to 80% cheaper than similar products developed in the west[33,34]. Given this advantage, the cost for selection of AEC is particularly favorable for China. The SOEC system is the least developed and has not yet been extensively commercialized. With extra heat input and an associated increase in working temperature, the electrical efficiency of SOEC is expected to be higher than that of alternatives. The data on the capital cost and electrical efficiency were obtained from projections by the International Energy Agency (IEA)[7] and Bloomberg New Energy Finance[34].

Electrolysis requires freshwater in addition to electricity. To produce 1 kg of hydrogen, 9 kg of freshwater are needed as feedstock. The actual freshwater consumption is higher due to evaporative loss, and we assumed that 1 kg of hydrogen requires 10 kg of freshwater[35]. In coastal areas, seawater desalination could provide freshwater sources for electrolysis. The desalination technology is mature in China with an installed capacity close to 800 Mt year$^{-1}$ in 2020[36]. With the increasing deployment of capacities for seawater desalination, the levelized cost for desalinated seawater has decreased in China to less than $1 m$^{-3}$ (ref. [37]). We assumed that the demand for freshwater could be supplied by desalination of seawater[38], and the cost for desalination was considered in the cost estimation. The annual consumption of freshwater through electrolysis is ~200 Mt for producing 20 Mt of hydrogen used for export to Japan. This would represent only 0.1% of total freshwater consumption in coastal China[37].

In the electrolysis process, the production of 1 kg hydrogen is associated with the by-production of 8 kg oxygen which could be sold and used both for industrial and medical purposes. The market for oxygen in China amounted to about 80 Mt in 2018, according to data reported in the Future Market Insights[39]. They projected that the market is likely to grow to 2030 at an annual rate of 6%, implying an annual market of about 160 Mt for oxygen by that time. Assuming a similar growth rate from 2030 to 2050, the annual demand could increase to about 500 Mt by the end of this period. The sources of oxygen estimated as byproducts of green hydrogen discussed here, 24 Mt and 160 Mt in 2030 and 2050, respectively, are relatively minor compared with the projected overall demand. The profit realized by selling the by-product oxygen, estimated at $40 ton$^{-1}$, was taken into account in the cost calculation for hydrogen[40]. Note that the value of $40 ton$^{-1}$ is chosen based on the cost for oxygen production by the current air separation process in China[41]. A further advantage to the green hydrogen-related production of oxygen is that the source in this case would be carbon-free.

**Hydrogen conversion**. Liquefaction, chemical bonding to toluene (forming MCH), and synthesis of ammonia were considered as options for subsequent

transport of hydrogen from its primary source to Japan. A low-pressure compressor (20 to 120 bar) is required to facilitate storage before the conversion of hydrogen. The techno-economic characteristics of hydrogen conversion technologies are summarized in Supplementary Table 2. Constraints on load level and ramping for the operation of conversion plants were formulated considering the impact of wind intermittency[42]. Capacities for individual plants indicated by the US Department of Energy (DOE) multi-year study[43] and the Japanese report from New Energy and Industrial Technology Development Organization (NEDO)[44] were adopted to scale the capital investment of each technology.

Theoretically, cooling 1 kg of hydrogen to its boiling point requires theoretically 3.9 k Wh of energy. In practice with current technology liquefaction of hydrogen involves close to 12 kWh kg$^{-1}$. Significant reductions in energy consumption and costs are anticipated with higher capacities for future systems[43]. For ammonia synthesis, nitrogen is supplied using air separation units, the electricity consumption of which leads to a slightly higher power demand for ammonia synthesis than the formation of MCH[44]. The material costs including carrier cost and catalyst cost were derived from the studies by Niermann et al.[12] and Dias et al.[45].

After transport to Japan, hydrogen in MCH and ammonia can be released through dehydrogenation for further use. As the reverse to hydrogenation, the dehydrogenation process is endothermic and requires inputs of heat energy in addition to electricity. In this context, we considered three possible sources of heat: one produced from hydrogen or hydrogen equivalent, one from the combustion of natural gas, and the other, when available, from the more economically advantaged deployment of waste heat supplied from combined heat and power plants. Future cost reductions and energy requirements of hydrogenation and dehydrogenation technologies were consistent with the estimations by the NEDO[44].

**Storage system**. Due to the variability of offshore wind power, storage facilities, are necessary to account for the fluctuations in hydrogen supply. In this study, gaseous hydrogen storage after hydrogen production, carrier storage after hydrogen conversion and buffer storage (7 days) at the port prior to the overseas transportation are the three types of storage facilities considered in the supply chain. The techno-economic characteristics of the storage technology options are summarized in Supplementary Table 3.

Pressurized tanks and geological reservoirs (salt caverns) are the two options considered for the storage of gaseous hydrogen. Since the geological option is geographically limited, pressurized tanks are taken as the default option for temporary storage after hydrogen production. Geological storage offers the possibility of a cost-effective hydrogen storage option. There are several existing salt caverns used for hydrogen storage in Europe and the United States. Qiu et al.[46] and Liu et al.[47] estimated the feasibility of large-scale underground hydrogen storage in Jiangsu province China. Additional calculations were conducted to evaluate the impacts of potential geological storage options in the coastal provinces of China (see Supplementary Fig. 2). The pressure for tank and cavern storage is assumed to be 120 bar, consistent with the outlet pressure of the compressor. The techno-economic data for hydrogen storage (including liquid hydrogen) were adopted from the US DOE report[43]. We assumed that the costs of salt caverns in the US DOE report do not include costs for compressors, and thus an additional low-pressure compression system was considered in our analysis to account for the related capital and electricity costs.

Bulk storage of ammonia is common in the industry. In liquid form, the product could be stored in conventional crude oils tanks, which is also a mature technology. Thus, related costs obtained from the national tank outlet[48] and the study by Niermann et al.[12] were assumed to be unchanged in 2030 and 2050. Comparatively speaking, the capital costs for transport of ammonia and MCH are much lower than those for alternative pressurized tanks or for the tanks that would be required to accommodate liquid hydrogen. This allows for the buildup of a significant reservoir compensating for times when the wind-fueled source may be temporally low. During the storage interval, we accounted also for anticipated boil-off from liquid hydrogen. The loss rates of all the technologies involved in the hydrogen supply chain were derived from the NEDO study[44].

**Hydrogen transport**. The concept of transporting liquid hydrogen by sea has already been considered. The world's first liquid hydrogen carrier ship was developed recently by Kawasaki Heavy Industries (KHI), unveiled in Kobe, Japan, in 2019[17]. MCH, like other liquid organic hydrogen carriers (LOHCs), has diesel-like attributes. It can be transported in existing oil tankers without the need for significant changes in these conveyances. For ammonia, the transportation and storage infrastructure already exists (more than 18 million tons of ammonia are traded globally annually)[49], eliminating the need for further development. Costs for the shipping options considered here were derived primarily from the study supported by NEDO[44]. For loading and unloading at ports, pump costs and energy consumption were specified following refs. [43,50,51]. Oversea trip distances and times were obtained from [52], which reports regular shipping information for international transportation. The techno-economic characteristics of the transportation technology options are summarized in Supplementary Table 4.

One of the reviewers to our paper (Dr. Ad van Wijk) raised the possibility of supplying the hydrogen to Japan using an underwater pipeline with potential advantages in avoiding several of the conversion processes[53]. We would point out,

though, that the Chinese hydrogen, produced most likely from a variety of spatially separated electrolyzers would have to be delivered to the underwater pipeline that might be constructed to facilitate the transport of hydrogen between the two countries. Costs for this delivery would need to be included in a detailed assessment of the pipeline idea. The assessment of such an option could be usefully followed up in a subsequent paper.

**Optimization model for least-cost hydrogen delivery**. The potential costs and related quantities of hydrogen that could be supplied to Japan are obtained based on a least-cost hydrogen delivery model optimizing jointly investment decisions and hourly operations accounting for the volatility of offshore wind power. The model (written using Release 2020b of the MATLAB and Simulink product families) allows for grid-by-grid analysis of offshore wind deployment, for the operation of electrolysis equipment, for hydrogen conversion, for energy storage, and for related overseas transportation. The mathematical formulation of the proposed optimization model is detailed introduced in the Supplementary Information.

**Reporting summary**. Further information on research design is available in the Nature Research Reporting Summary linked to this article.

## Data availability
Wind resources data are publicly available from NASA's Goddard Earth Observing System database. All other data are available in the manuscript and the Supplementary Information.

## Code availability
Code for the optimization model can be made available upon request.

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

## Acknowledgements

We thank Y. Shibata, S. Kan, Y. Zhang, and M. Nagashima for helpful discussions. This work was supported by Ningxia Yanbao Charity Foundation (Grant No. G-2021-1), Harvard Global Institute, Harvard President's Office, Energy Foundation China, and Sze Family Foundation. H.L. acknowledges the support from the program for an Outstanding Ph.D. candidate of Shandong University. X.Y. acknowledges the National Natural Science Foundation of China (Grant No. 71704187) and Horizontal 2020 European Commission Project "PARIS REINFORCE" (Grant No. 820846).

## Author contributions

M.B.M., S.S. and C.P.N. conceived and designed the project; S.S., H.L., P.S. and X.C. collected data and performed the calculation; S.S., H.L., P.S., M.B.M. and X.Y. analyzed the data; S.S., H.L. and M.B.M. wrote the manuscript, with P.S., X.Y., C.P.N. and X.C. contributing to its final version. These authors contributed equally: S.S. and H.L.

## Competing interests

The authors declare no competing interests.
