## [Peer Review File · Nature Communications]

Peer Review comments, first round review –

Reviewer #1 (Remarks to the Author):

Thank you for the opportunity to review this manuscript.

While I found the contents interesting, and indeed the potential for a low-cost pathway from China appears to exist, the analysis is not novel, nor is it distinct from a number of pathways already explored in the literature, i.e. the low-carbon import of H₂ from Australia and comparative pathways, and South American pathways. I feel that this work would be better placed in a specific energy or hydrogen journal which accepts case studies.

Reviewer #2 (Remarks to the Author):

The authors investigated the production capacity and cost of Chinese green hydrogen which will export to Japan. They focus on the offshore wind farms as a power source for water electrolysis and considered three types of carriers (liquid hydrogen, MCH and ammonia) for ocean transportation of hydrogen energy. They found that the estimated production capacity can be greater than the future hydrogen demand in Japan. It is also reported that levelized costs of hydrogen can be lower than the target cost of Japan (\$3.0/kg in 2030 and \$2.0/kg in 2050) if appropriate supply chains are selected. The methods used in this study is clear. The data is clearly presented, and the obtained results will be of use to researchers and policy makers in the field.

I would recommend it for acceptance after the minor points listed below:

Supplementary Information

#1 (Page 2): C_wat, γ _wat and r_wat in (S3) should be revised to C_wt, γ _wt and r_wt, respectively.

#2 (Page 2): Please explain the definition of O&M fractions, which will be helpful for readers who are not familiar with cost analysis.

#3 (Page 3): Γ _wt and R_wt should be changed to small characters.

#4 (Page 8): PEMAEC should be revised to PEMEC

#5 (Page 14): Please explain why the unit capital cost of compression in 2050 will be higher than that in 2030.

#6 (Page 14): Techno-economic data of liquefaction and ammonia cracking are both written as "2030".

Reviewer #3 (Remarks to the Author):

The paper analyses the hydrogen potential as a function of the LCoH for hydrogen produced from offshore wind in China, shipped to Japan as liquid hydrogen, ammonia or bound to toluene. The analyses show clearly that Japan's targets for 2050, 20 Mton hydrogen consumption at a cost below \$2/kg H₂ could be fully met by China's offshore wind hydrogen production. These conclusions clearly show that hydrogen can become the global energy commodity, with which cheap renewable electricity from solar and wind can be transported cost-effectively around the world. This study is therefore a valuable contribution to the scientific literature analysing the role of hydrogen in a future sustainable energy system.

The methodology used is clearly described using 30 year wind speed data to calculate electricity production at all offshore areas with water depths less than 60 meter. By electrolysis of water, hydrogen is produced, which is converted into liquid hydrogen, ammonia or bound to toluene and shipped to Japan, to be converted back to gaseous hydrogen. Figure S1 shows the system design for the supply chain, whereby for all elements in the supply chain relevant input data about Capex, Opex, lifetime, efficiencies and energy input and cost are analysed and showed in tables. Although clear, I want to address some issues about the calculation methodology, system design, chosen technologies and want to make some remarks about the input data.

In the LCoH calculations, the income by selling the Oxygen that is produced too by electrolysis of water is subtracted from the cost. With every kg of hydrogen also 8 kg of oxygen is produced and sold at \$40 per ton oxygen. So this reduces the LCoH with \$0.32/kg H₂. But it is very doubtful that there is a market for such large quantities of hydrogen, 20 Mton hydrogen means 160 Mton oxygen. Could the authors indicate what the market for oxygen is and what the effect on the LCoH is when not all produced oxygen can be sold or at a lower price. Besides what are the associated cost to store and transport oxygen to the demand and how does this affect the sales price?

The wind turbine characteristics that are used in this study, are from the Vestas Offshore wind turbine V164-8.0 MW. However, today larger offshore wind turbines are on the market, with larger rotor diameters and higher rated power, for example the Vestas V236-15MW offshore wind turbine, showing also higher capacity factors. Could these wind turbines be applied in Chinese offshore conditions? And if so what would be the effect on the LCoH calculations?

In this study the electricity produced by wind turbines is transported by electricity cable to the shore, where a centralized large scale electrolyser on land is placed to produce hydrogen (if I am right). However, today several companies, among others Siemens, investigate to install an electrolyser in or at the offshore wind turbine. It offers possible advantages to reduce electricity conversion cost and cheaper pipeline transport cost. Could this new system technology for integrated offshore wind hydrogen production be an option at least for 2050? And how could this effect the LCoH?

The transport distances by ship from China to Japan Kobe vary between 1,300 and 3,300 km, see table S5. But for these distances also hydrogen transport by pipeline can be considered. 48 inch diameter pipelines have a transport capacity between 15 and 20 GW depending on operating pressure and flow speed, can transport 2-2.5 Mton hydrogen per year at 5,000 full load hours and show hydrogen transport cost of \$ 0.15-0.25 per kg H₂ per 1,000 km. Although transport by ship is more flexible, transport cost by pipeline is probably cheaper and avoids several conversion processes. Pipeline transport in 2050, immediately from the offshore wind hydrogen production locations, could possible be a feasible and cost competitive option. Could this be an option for hydrogen transport from China to Japan? And what would be the effect on the LCoH?

In table S2 techno-economic data for compression are given. But it is not clear to me where the compression takes place, it is not indicated in figure S1, is it for storage in tanks or salt caverns? What is the required pressure and what is the input pressure? It seems also that the compression input data for 2030 and 2050 have been swapped? And the figure for electricity demand in 2030 and 2050 differ a lot, which seems strange, if in 2030 and 2050 the same pressure difference has to be realized by compression. And maybe I missed it, but I do not see an explanation about the scale factor and where this is used.

In table S2 also the input data for liquefaction are given. The electricity demand for liquefaction is 6.76 kWh/kg H₂ in 2030, going down to 1.51 kWh/kg H₂ in 2050. But the thermodynamic minimum energy needed to liquify hydrogen from ambient temperature gaseous hydrogen is 3.9 kWh/kg H₂. How could this figure for 2050 in the table be explained?

In table S3, the input data for salt caverns are given. The Capital cost seems to be rather low, do this

cost figure include leaching the salt cavern, disposal of brine, compressor cost, hydrogen cleaning cost, etc? Next to these cost hydrogen 'cushion' gas has to be injected to build up a baseload pressure in the salt cavern (60-80 bar), before you can store hydrogen with pressures up to 200 bar. Could these salt cavern input data be elaborated?

Prof. dr. Ad van Wijk

Response to Editorial Requests

Reply: In addition to the responses to the three reviewers and the corresponding revisions in the manuscript and supplementary material, we have also revised the format of the manuscript following the formatting instructions

Response to Reviewers

Reviewer #1 (Remarks to the Author):

Thank you for the opportunity to review this manuscript. While I found the contents interesting, and indeed the potential for a low-cost pathway from China appears to exist, the analysis is not novel, nor is it distinct from a number of pathways already explored in the literature, i.e. the low-carbon import of H₂ from Australia and comparative pathways, and South American pathways. I feel that this work would be better placed in a specific energy or hydrogen journal which accepts case studies.

Reply: A primary objective of our study is to explore the quantity and cost of green hydrogen that could be produced from offshore wind in China and delivered to Japan. As we noted in the introduction, there are obviously potential other sources of hydrogen, either green or blue, that could be supplied from other countries, but the costs of the imported hydrogen could be higher than the targets Japan has identified to sustain a viable future hydrogen economy. For the first time, our paper combines multiple-year wind resource data from a state-of-the-art meteorological dataset and a comprehensive technical-economic supply chain analysis to investigate the prospects of green hydrogen from China. The key conclusion here is that the potential Chinese source could meet Japan's large term objectives and could do so at a cost lower than those quoted for the other countries. We appreciate the reviewer's generally positive reactions to the paper. We do believe though that *Nature Communications* represents a suitable journal for our paper.

Reviewer #2 (Remarks to the Author):

The authors investigated the production capacity and cost of Chinese green hydrogen which will export to Japan. They focus on the offshore wind farms as a power source for water electrolysis and considered three types of carriers (liquid hydrogen, MCH and ammonia) for ocean transportation of hydrogen energy. They found that the estimated production capacity can be greater than the future hydrogen demand in Japan. It is also reported that leveled costs of hydrogen can be lower than the target cost of Japan (\$3.0/kg in 2030 and \$2.0/kg in 2050) if appropriate supply chains are selected. The methods used in this study is clear. The data is clearly presented, and the obtained results will be of use to researchers and policy makers in the field. I would recommend it for acceptance after the minor points listed below:

Reply: We thank the reviewer's positive comments on our manuscript and have made changes for these points in the revised manuscript and supplementary material.

Supplementary Information

#1 (Page 2): C_wat , γ_wat and r_wat in (S3) should be revised to C_wt , γ_wt and r_wt , respectively.

Reply: These symbols have been corrected.

#2 (Page 2): Please explain the definition of O&M fractions, which will be helpful for readers who are not familiar with cost analysis.

Reply: Thanks for pointing it out. We have added the full expression of O&M, i.e., Operations and Maintenance, when this term is introduced in the manuscript.

#3 (Page 3): Γ_wt and R_wt shoed be changed to small characters.

Reply: These symbols have been corrected.

#4 (Page 8): PEMAEC should be revised to PEMEC

Reply: Thanks for pointing this typo out. It has been corrected.

#5 (Page 14): Please explain why the unit capital cost of compression in 2050 will be higher than that in 2030.

Reply: We thank the reviewer for pointing out the issues in the techno-economic data in Table S2. There have been typos in Table S2 of the original manuscript, which have been corrected in the revised one. The figure we assume for the unit capital cost of compression in 2050 is $0.36 \text{ k\$ kg}^{-1} \text{ h}^{-1}$, which is projected to be lower than that of $0.41 \text{ k\$ kg}^{-1} \text{ h}^{-1}$ in 2030.

#6 (Page 14): Techno-economic data of liquefaction and ammonia cracking are both written as "2030".

Reply: Thanks for pointing these typos out. They have been corrected.

Reviewer #3 Prof. Dr. Ad van Wijk (Remarks to the Author):

The paper analyses the hydrogen potential as a function of the LCoH for hydrogen produced from offshore wind in China, shipped to Japan as liquid hydrogen, ammonia or bound to toluene. The analyses show clearly that Japan's targets for 2050, 20 Mton hydrogen consumption at a cost below \$2/kg H₂ could be fully met by China's offshore wind hydrogen production. These conclusions clearly show that hydrogen can become the global energy commodity, with which cheap renewable electricity from solar and wind can be transported cost-effectively around the world. This study is therefore a valuable contribution to the scientific literature analysing the role of hydrogen in a future sustainable energy system.

The methodology used is clearly described using 30 year wind speed data to calculate electricity production at all offshore areas with water depths less than 60 meter. By electrolysis of water, hydrogen is produced, which is converted into liquid hydrogen, ammonia or bound to toluene and shipped to Japan, to be converted back to gaseous hydrogen. Figure SI shows the system design for the supply chain, whereby for all elements in the supply chain relevant input data about Capex, Opex, lifetime, efficiencies and energy input and cost are analysed and showed in tables. Although clear, I want to address some issues about the calculation methodology, system design, chosen technologies and want to make some remarks about the input data.

Reply: We acknowledge Prof. Dr. Ad van Wijk's positive comments on the significance and conclusion of our study. We totally agree with the reviewer that green hydrogen can become an important global zero-carbon energy commodity in the future. We also appreciate the positive comments on the methodology that we use in this study to evaluate the potential supply chain of green hydrogen from China to Japan.

We thank the reviewer for constructive and detailed comments and suggestions for improvements in relation to the calculation methods, chosen technologies, and input data which have prompted, we believe, significant improvements in our paper. Below, we provide point-by-point responses to specific comments and suggestions noting changes have been made in the revised manuscript and supplementary material.

In the LCoH calculations, the income by selling the Oxygen that is produced too by electrolysis of water is subtracted from the cost. With every kg of hydrogen also 8 kg of oxygen is produced and sold at \$40 per ton oxygen. So this reduces the LCoH with \$0.32/kg H₂. But it is very doubtful that there is a market for such large quantities of hydrogen, 20 Mton hydrogen means 160 Mton oxygen. Could the authors indicate what the market for oxygen is and what the effect on the LCoH is when not all produced oxygen can be sold or at a lower price. Besides what are the associated cost to store and transport oxygen to the demand and how does this affect the sales price?

Reply: We thank the reviewer for this important comment. According to the *Industrial Oxygen Market* report by Future Market Insights (FMI), the market for oxygen in China was about 80 Mton in 2018. The oxygen market is projected by FMI to grow at an annual rate of about 6% to around 2030 driven by the increasing demand in the healthcare, construction, and aerospace sectors. This translates to a Chinese oxygen market of about 160 Mton in 2030 and about 500 Mton in 2050 assuming the same growth rate from 2030 to 2050. The projected quantities of hydrogen shipped to Japan are 3 Mton and 20 Mton respectively for 2030 and 2050, which correspond to 24 Mton and 160 Mton oxygen for 2030 and 2050, respectively. Therefore, a rough estimate is that the co-produced oxygen represents a minor contribution to the projected future Chinese oxygen market.

We estimate that the LCoH is reduced by $\$0.32 \text{ kg}^{-1}$ assuming an economic benefit of $\$40 \text{ ton}^{-1}$ for the co-produced oxygen. This value of $\$40 \text{ ton}^{-1}$ is chosen based on the current production cost of oxygen from the current air separation process. The selling price of industrial oxygen in the Chinese market is about $\$100 \text{ ton}^{-1}$.

We also note that an additional advantage to the green hydrogen related production of oxygen is the fact that this source would be carbon free. The air separation process, which is commonly used to produce oxygen in China, leads to significant carbon emissions since the current consumed electricity is generated largely by coal.

We have modified the manuscript as indicated to summarize these considerations. Please see Lines 337-349.

The wind turbine characteristics that are used in this study, are from the Vestas Offshore wind turbine V164-8.0 MW. However, today larger offshore wind turbines are on the market, with larger rotor diameters and higher rated power, for example the Vestas V236-15MW offshore wind turbine, showing also higher capacity factors. Could these wind turbines be applied in Chinese offshore conditions? And if so what would be the effect on the LCoH calculations?

Reply: We thank the reviewer for this important comment. It is true that offshore wind turbines with higher rated power are currently being introduced into the market and could represent a major cost reduction for green hydrogen production. Unfortunately, there are not enough data publicly available for the 15 MW turbine to define the relevant power curve. We have thus included in the revised paper a study of the potential for applications of the 9.5 MW Vestas turbine substituting for the 8.0 MW Vestas option considered here. We find that the yield of electricity that could be realized using the 9.5 MW turbine could be greater than the 8.0 MW case by roughly 19%. Assessment of cost potentials for production of hydrogen using the 9.5 MW option would require however data on the relative costs of these two options. It is additionally difficult to estimate the economics that might be associated with the even larger Vestas V236-15 MW option, lacking even information on the relevant power curve. Given the larger diameter of this turbine, it would be necessary in this case to impose a greater separation between individual turbines relative to either the 8.0 MW or 9.5 MW options.

In the revised paper, we have included additional text acknowledging the potential for larger turbines in the future noting the importance of a treatment for the relevant economics, as these turbines begin to be introduced into the rapidly expanding offshore wind market. Please see Lines 271-279.

In this study the electricity produced by wind turbines is transported by electricity cable to the shore, where a centralized large scale electrolyser on land is placed to produce hydrogen (if I am right). However, today several companies, among others Siemens, investigate to install an electrolyser in or at the offshore wind turbine. It offers possible advantages to reduce electricity conversion cost and cheaper pipeline transport cost. Could this new system technology for integrated offshore wind hydrogen production be an option at least for 2050? And how could this effect the LCoH?

Reply: Yes. In our study, it is assumed that the electricity produced by offshore wind turbines is transported by electricity cable to the shore, where a centralized large-scale electrolyzer on land is installed to produce green hydrogen. The reviewer raises an interesting point that production of hydrogen could be sited not onshore but at the wind farm itself. We would argue, though, that the primary product of offshore Chinese wind installations will be to deliver electricity to the

coastal market and that the hydrogen for Japan explored here represents a minor application of this power. We certainly agree that if the primary purpose of an offshore wind farm is to produce hydrogen, the possibilities represented by products such as those considered by Siemens could serve as an important future market. We have modified the manuscript as indicated to summarize these considerations. Please see Lines 97-103.

The transport distances by ship from China to Japan Kobe vary between 1,300 and 3,300 km, see table S5. But for these distances also hydrogen transport by pipeline can be considered. 48 inch diameter pipelines have a transport capacity between 15 and 20 GW depending on operating pressure and flow speed, can transport 2-2.5 Mton hydrogen per year at 5,000 full load hours and show hydrogen transport cost of \$ 0.15-0.25 per kg H₂ per 1,000 km. Although transport by ship is more flexible, transport cost by pipeline is probably cheaper and avoids several conversion processes. Pipeline transport in 2050, immediately from the offshore wind hydrogen production locations, could possible be a feasible and cost competitive option. Could this be an option for hydrogen transport from China to Japan? And what would be the effect on the LCoH?

Reply: The reviewer raises an interesting possibility, that hydrogen from the Chinese source could be delivered to Japan by pipeline rather than ship. The economics for such a possibility certainly merit further attention. We are aware of the reviewer's proposal to transport hydrogen from Africa to Europe by a pipeline system. We would point out, though, that the hydrogen delivery by such a system in China would need to be supplied from a variety of spatially separated onshore sources. Costs for this supply would have to be added to the expense envisaged for a possible underwater pipeline system linking the two countries China and Japan. We have modified the manuscript as indicated to summarize these considerations. Please see Lines 414-421.

In table S2 techno-economic data for compression are given. But it is not clear to me where the compression takes place, it is not indicated in figure S1, is it for storage in tanks or salt caverns? What is the required pressure and what is the input pressure? It seems also that the compression input data for 2030 and 2050 have been swapped? And the figure for electricity demand in 2030 and 2050 differ a lot, which seems strange, if in 2030 and 2050 the same pressure difference has to be realized by compression. And maybe I missed it, but I do not see an explanation about the scale factor and where this is used.

Reply: We thank the reviewer for this important comment. The compression takes place after the water electrolysis and before the gas hydrogen storage. In the revised supplementary material, we have modified Figure S1 to indicate this process in the hydrogen supply chain scheme. In the analysis, we assume that a low-pressure compressor (20 bar to 120 bar) is equipped to facilitate tank and geological storage. This has been clarified in the revised manuscript.

We thank the reviewer for pointing out the issues with the techno-economic data in Table S2. There have been typos in Table S2 of the original manuscript, which have been corrected in the revised one. The figures for electricity demand in 2030 and 2050 are the same, 0.84 kWh kg⁻¹, since the same pressures are assumed for the compression process.

The scale factor listed is not used in this study since the subsystem capacity exceeds the capacity limitation of each unit. The numbers might be used by readers to build a smaller system, so we chose to show it to provide a reference. A note has been added to explain this at the bottom of Table S2 in the revised supplementary material (Page 14): “Scale factor is only used when designed system capacity is smaller than unit capacity”.

In table S2 also the input data for liquefaction are given. The electricity demand for liquefaction is 6.76 kWh/kg H₂ in 2030, going down to 1.51 kWh/kg H₂ in 2050. But the thermodynamic minimum energy needed to liquify hydrogen from ambient temperature gaseous hydrogen is 3.9 kWh/kg H₂. How could this figure for 2050 in the table be explained?

Reply: We thank the reviewer for pointing this out. Similar to the data issue for compression, there have been typos in Table S2 of the original manuscript, which have been corrected in the revised one. The electricity demand for liquefaction we assume in 2050 is 6 kWh kg⁻¹ H₂, which is larger than the thermodynamic minimum energy (3.9 kWh kg⁻¹ H₂) required to liquify hydrogen from ambient temperature gaseous hydrogen.

In table S3, the input data for salt caverns are given. The Capital cost seems to be rather low, do this cost figure include leaching the salt cavern, disposal of brine, compressor cost, hydrogen cleaning cost, etc? Next to these cost hydrogen ‘cushion’ gas has to be injected to build up a baseload pressure in the salt cavern (60-80 bar), before you can store hydrogen with pressures up to 200 bar. Could these salt cavern input data be elaborated?

Reply: We thank the reviewer for this important comment. In this study, we did not include a detailed estimation for salt cavern storage costs based on the China context, and the capital costs for salt caverns we use here are adopted from a technical report by the U.S. Department of Energy (DOE) which is cited in the manuscript. The costs in the U.S. DOE report are slightly lower than the calculation results from HDSAM V3.0 designed by Argonne, which includes the costs for cavern disposal, compressor, other necessary equipment, and cushion gas. We assume that the U.S. DOE salt cavern numbers do not involve compressors, which is consistent with the tank storage option, and thus an additional low-pressure compression system is considered in our analysis to represent the related capital and electricity costs. The capital cost in 2015 estimated by the U.S. DOE study is \$16 kg⁻¹, and reductions in this cost are projected to 2030 and 2050 (values shown in Table S3). We estimate, if \$16 kg⁻¹ is taken as the salt cavern capital cost, that the LCoH would increase by about 5% compared to current assumptions but is still lower compared to the case using pressurized tanks. We have modified the manuscript as indicated to elaborate the input data for the salt cavern storage. Please see Lines 389-392.

Peer Review comments, additional round review –

Reviewer #1 (Remarks to the Editor):

The manuscript has been much improved and is in nice condition now.

No further review comment/s.